

# Effect of body mass index on soft tissues in adolescents with skeletal class I and normal facial height

Nurver Karsli and Esra Tuhan Kutlu

Department of Orthodontics, Faculty of Dentistry, Karadeniz Technical University, Trabzon, Turkey

## ABSTRACT

**Background:** The evaluation of soft tissues in patients undergoing orthodontic treatment plays a critical role in diagnosis and treatment planning. This study aims to evaluate the effect of body mass index (BMI) on facial soft tissues in skeletal Class I patients with normal vertical growth.

**Methods:** The study included 72 patients with a normal vertical growth pattern (SN-GoGn 26–38°) and skeletal Class I (ANB 2–4°) malocclusion. According to their BMI (kg/m$^2$) values, the patients were divided into three groups of 24 individuals each: underweight (>5th percentile) (13 females, 11 males; mean age, 11.58 ± 1.95 years), normal weight (5–85th percentile) (12 females, 12 males; mean age, 11.54 ± 1.95 years), overweight (85–95th percentile) (12 females, 12 males; mean age, 11.62 ± 2.01 years). Soft tissue thickness and height measurements were made on lateral cephalometric radiographs.

**Results:** In all soft tissue thickness measurements, except for the nasion, the overweight weight group had higher values. In comparing the underweight and normal weight groups, statistically significant differences were found in the thickness measurements at the nasion and gnathion ($p < 0.05$). The differences in thickness measurements at the glabella, labiale superius, stomion, labiale inferius, pogonion, gnathion, and menton are statistically significant between the underweight and overweight groups ($p < 0.005$). Comparing the normal and overweight groups revealed statistically significant differences the thickness measurements at the glabella, labiale superius, stomion, pogonion and menton ($p < 0.05$).

## INTRODUCTION

Although growth and development are closely related concepts, they are not synonymous. Growth is mostly an anatomical condition an increase in anatomical dimensions. It involves an increase in the number and size of cells in an organism and represents an increase in body volume and mass (*Ferguson & Dean, 2015*). On the other hand, development is both physiological and behavioral, signifying an increase in the specialization and organization of cell functions (*Proffit & Fields, 2007*).

Many factors, such as genetics, gender, nutrition, physical activity, and socioeconomic status, affect growth. Nutrition is one of the most important factors influencing growth in

Corresponding author
Nurver Karsli,
dtnurverkarsli@hotmail.com

the first 2 years of life (*Yağcı, 2005*). Although malnutrition includes both overnutrition and undernutrition, it is generally accepted as a nutritional deficiency (*Shaughnessy & Kirkland, 2016*). Obesity is a common nutritional disorder in children (*Elkum et al., 2016*). It is the abnormal and excessive accumulation of fat in adipose tissues and other organs that impairs health. It has been reported that obesity accelerates skeletal maturation and growth in childhood and before puberty (*Freedman et al., 2003*).

Many measurement methods, such as waist circumference, skinfold thickness, and body mass index (BMI), have been used to assess the weight status of individuals (*Mack et al., 2013*). BMI has a significant effect on teeth and skeletal development. It has been proven that overweight and obese children are more skeletally developed than their peers with a normal BMI (*DuPlessis et al., 2016*).

The evaluation of soft tissues in patients undergoing orthodontic treatment plays a critical role in diagnosis and treatment planning (*Kamak & Celikoglu, 2012*; *Celikoglu et al., 2015*). Both hard and soft tissue norms should be considered to achieve facial esthetic harmony and optimal functional occlusion. Patients with thick and thin soft tissues have been reported to have different soft tissue thickness changes after bimaxillary surgery (*Abeltins & Jakobsone, 2011*).

We aimed to investigate the relationship between BMI and facial soft tissue thickness and height in patients with continuing growth and development. This study will aid clinicians in predicting the soft-tissue thicknesses of patients based on body mass index, before patient documentation or when it is not desired to take radiographs for radiation reduction purposes.

## MATERIALS AND METHODS

The ethics committee of the Faculty of Dentistry Karadeniz Technical University approved this study (number: 2022/4, date: 07/09/2022). Participants were selected from patients who consulted to the university orthodontic clinic. Informed consent was obtained from the patients participating in the study. Before the study, the number of patients required for statistically significant data was determined with the power analysis in light of similar previous studies (*Celikoglu et al., 2015*). The study included 72 (35 male and 37 female) patients who were receiving orthodontic treatment and had a normal vertical growth pattern (SN-GoGn 26–38°) and skeletal Class I (ANB 2–4°) malocclusion. Patients with a previous history of orthodontic treatment or orthognathic surgery, those with a history of head trauma, and those with congenital anomalies and syndromes were excluded.

The participants' height was measured in the clinic using a height ruler, while their weight was measured with a digital scale. While performing the height measurement, it was ensured that each patient was not wearing shoes, and the heel, hip, shoulder, and back of the head formed a Frankfurt plane, with the eye-ear plane positioned perpendicular to the wall. During weight measurements, care was taken to ensure that the patient wore thin clothes. The BMI values of the patients were calculated by dividing their weight (kg) by the square of their height (m). According to the percentile values in the BMI (kg/m$^2$) charts defined according to age and gender published by the World Health Organization (WHO), the patients were divided into the following three groups of 24 individuals each:

underweight (>5th percentile) (13 females, 11 males; mean age, 11.58 ± 1.95 years), normal weight (5–85th percentile) (12 females, 12 males; mean age, 11.54 ± 1.95 years), overweight (85–95th percentile) (12 females, 12 males; mean age, 11.62 ± 2.01 years).

Lateral cephalometric radiographs were obtained using cephalometric filming device (Kodak 9000 Extraoral Imaging System, Cephalostat; Corestream Health Inc., Rochester NY, USA), while the head was in natural head position. The distance of subject to x-ray source/the mid-occlusal plane of film cassette was standardized for each patient as 119 and 13 cm, respectively and 0.364 s radiation at 73 kW, 15 mA was applied. Magnification of cephalograms was 5% and all linear measurements were corrected accordingly. The same technician took lateral cephalometric radiographs, placing each patient in a relaxed lip position with maximum intercuspation. Soft tissue thickness and height measurements were made on these radiographs by an experienced orthodontist using NemoCeph (Version 6.0; Nemotec, Madrid, Spain) software. Soft tissue thickness measurements evaluated: Glabella (G-G′), Nasion (N-N′), A point-Subnasale (A-Sn), Labiale superius (Pr-Ls), Stomion (ls-Sto), Labiale inferius (id-Li), Labiomentale (B-Lm), Pogonion (Pog-Pog′), Gnathion (Gn-Gn′), Menton (Me-Me′). Soft tissue height measurements evaluated: subnasale-upper labial bottom (Sn-ULB), lower lip bottom-lower lip vermillion (LLB-LLV), subnasale-lower lip vermillion (Sn-LLV).

Figures 1 and 2 present the patients' soft tissue thicknesses and height measurements. The measurements were repeated 2 weeks later to ensure accuracy by the same person.

## Statistical analysis

Statistical analyses were performed using SPSS for Windows v. 17.0. The conformity of the data to the normal distribution was evaluated using the Shapiro-Wilk test. Descriptive statistics were given as mean, standard deviation, or percent requency values (gender). The paired-sample t-test was used for inter-group changes, while a one-way analysis of variance and Tukey's *post hoc* test were used to compare the groups. The reliability of clinical measurements made by the observer at different times was evaluated by calculating the intraclass correlation coefficient (ICC) and 95% confidence intervals. ICC estimates and their 95% confident intervals were calculated based on absolute-agreement, two-way mixed-effects model. ICC level; If it was between 0.80 and 1.00, it was accepted that 1st and 2nd measurements were highly consistent. A *p*-value of <0.05 was considered significant in all analyses.

## RESULTS

Intraclass correlation coefficients for facial soft tissue thickness and height measurements are shown in Table 1. The demographic data of the patients included in the study groups are shown in Table 2. The groups were statistically well-matched in gender distribution, chronological age, vertical growth pattern, and sagittal relationships ($p > 0.05$). There were statistically significant differences between the groups in terms of the mean BMI values (underweight group, 16.54 ± 0.99; normal weight group, 19.80 ± 1.67; overweight group, 25.96 ± 2.13) ($p < 0.05$).

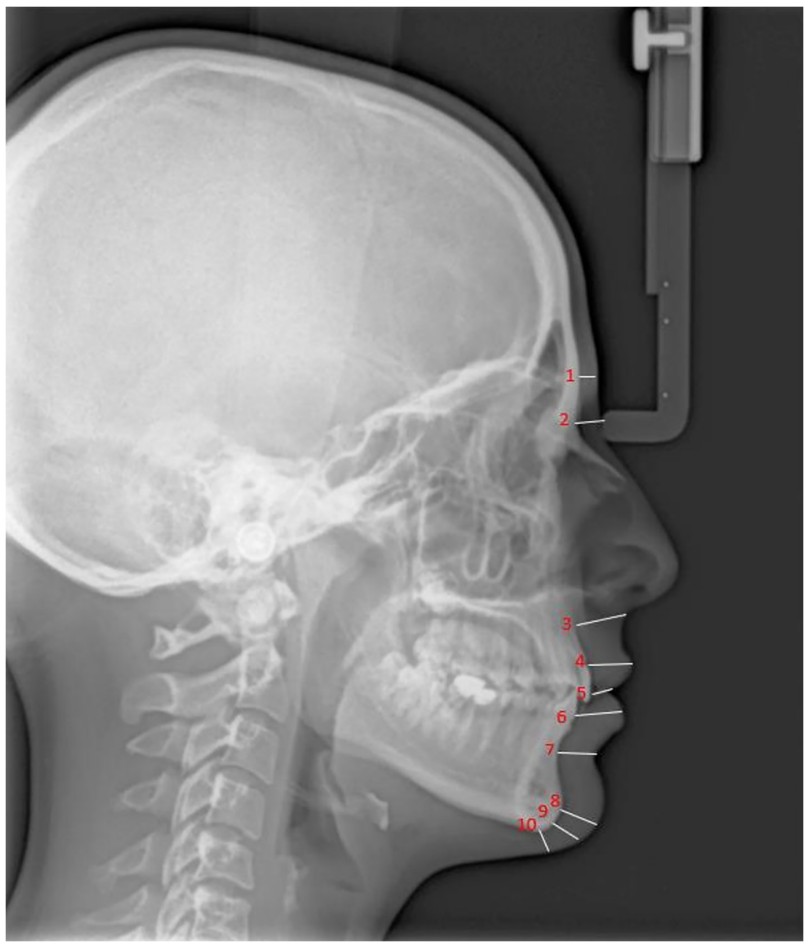

**Figure 1 Soft tissue thickness measurement.** 1, Glabella (G-G′); 2, Nasion (N-N′); 3, A point-Subnasale (A-Sn); 4, Labiale superius (Pr-Ls); 5, Stomion (ls-Sto); 6, Labiale inferius (id-Li); 7, Labiomentale (B-Lm); 8, Pogonion (Pog-Pog′); 9, Gnathion (Gn-Gn′); 10, Menton (Me-Me′); G, Hard tissue Glabella; G′, Soft tissue Glabella; N, Hard tissue Nasion; N′, Soft tissue Nasion; A, A Point; Sn, Subnasale; Pr, Prosthion; Ls, Labrale superius; ls, upper incisor edge; Sto, Stomion; id, infradentale; Li, Labrale inferius; B, B Point; Lm, Labiomentale; Pog, Hard tissue Pogonion; Pog′, Soft tissue Pogonion; Gn, Hard tissue Gnathion; Gn′, Soft tissue Gnathion; Me, Hard tissue Menton; Me′, Soft tissue Menton.

Table 3 shows the mean soft tissue thickness measurements of the BMI percentile groups and their inter-group comparisons. In all soft tissue thickness measurements, except for the nasion, the overweight group had higher values. The difference in thickness measurements at the nasion and gnathion between the underweight and normal weight groups is statistically significant ($p < 0.05$). When the underweight and overweight groups were compared, statistically significant differences were found with the thickness measurements at the glabella, labiale superius, stomion, labiale inferius, pogonion, gnathion, and menton ($p < 0.005$). Comparing the normal and overweight groups revealed statistically significant differences the thickness measurements at the glabella, labiale superius, stomion, pogonion and menton ($p < 0.05$).

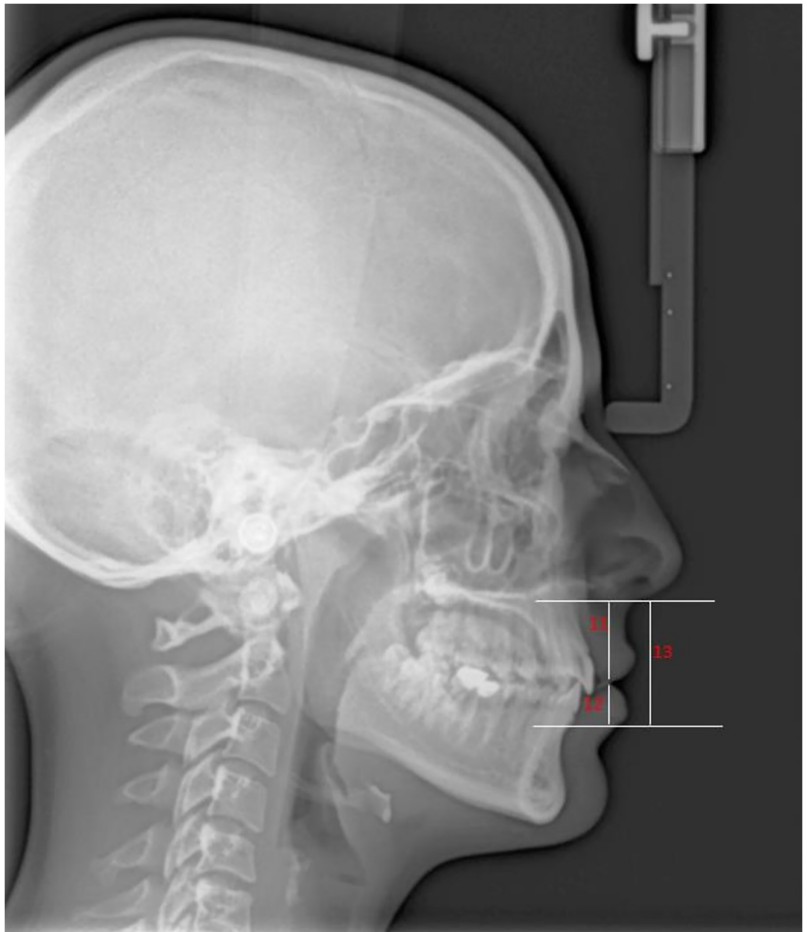

**Figure 2** **Soft tissue height measurement.** 11, Subnasale-Upper Labial Bottom (Sn-ULB); 12, Lower Lip Bottom-Lower Lip Vermillion (LLB-LLV); 13, Subnasale-Lower Lip Vermillion (Sn-LLV).

Table 4 shows the mean soft tissue height measurements of the BMI percentile groups and their inter-group comparisons. No statistically significant difference was found between the groups' height measurements ($p < 0.05$).

## DISCUSSION

Many factors, such as genetics, gender, nutrition, and socioeconomic status, affect growth and development (*Demirjian, Goldstein & Tanner, 1973*; *Demirjian & Goldstein, 1976*; *Rogol, Clark & Roemmich, 2000*; *Hilgers et al., 2006*; *Mehta et al., 2013*). Various studies have shown that nutrition is the most effective of these factors (*Yağcı, 2002*). In children with malnutrition, there may be changes in growth rate, sexual maturation, and skeletal and dental development (*Freedman et al., 2003*; *Hilgers et al., 2006*; *Slyper, 2006*; *Akridge et al., 2007*; *Vandewalle et al., 2014*).

Studies have shown that methods allowing for direct body fat measurements in evaluating the nutritional status, such as densitometry, ultrasonography, dual photon absorptiometry, computed tomography, and magnetic resonance imaging, yield precise

**Table 1 Intraclass correlation coefficients for facial soft tissue thickness and height measurements.**

|  | ICC | 95% CI | p |
|---|---|---|---|
| ANB | 0.856 | [0.751–0.918] | <0.001 |
| GoGn-SN | 0.958 | [0.926–0.977] | <0.001 |
| Glabella | 0.933 | [0.882–0.963] | <0.001 |
| Nasion | 0.890 | [0.809–0.938] | <0.001 |
| A point-subnasale | 0.958 | [0.925–0.976] | <0.001 |
| Labiale superius | 0.955 | [0.920–0.975] | <0.001 |
| Stomion | 0.944 | [0.892–0.970] | <0.001 |
| Labiale inferius | 0.964 | [0.935–0.980] | <0.001 |
| Labiomentale | 0.904 | [0.832–0.946] | <0.001 |
| Pogonion | 0.959 | [0.926–0.977] | <0.001 |
| Gnathion | 0.927 | [0.871–0.959] | <0.001 |
| Menton | 0.932 | [0.880–0.962] | <0.001 |
| Sn-ULB | 0.953 | [0.916–0.974] | <0.001 |
| LLB-LLV | 0.948 | [0.907–0.971] | <0.001 |
| Sn-LLV | 0.963 | [0.933–0.979] | <0.001 |

Note:
ICC, Intraclass Correlation Coefficient; CI, Confidence Interval.

**Table 2 Demographic data of the patients in different BMI percentile groups.**

|  | n | Female/male | Mean age (Years) | ANB (°) | SN-GoGn (°) | BMI (kg/m$^2$) | Percentile (%) |
|---|---|---|---|---|---|---|---|
| Underweight group | 24 | 13/11 | 11.58 ± 1.95 | 3.12 ± 0.89 | 31.83 ± 2.35 | 16.54 ± 0.99 | <5 |
| Normal weight group | 24 | 12/12 | 11.54 ± 1.95 | 3.20 ± 0.93 | 32.29 ± 2.45 | 19.80 ± 1.67 | 40 ± 19.71 |
| Overweight group | 24 | 12/12 | 11.62 ± 2.01 | 2.87 ± 0.85 | 31.45 ± 2.65 | 25.96 ± 2.13 | 88.45 ± 2.79 |
| Total | 72 | 37/35 | 11.58 ± 1.95 | 3.06 ± 0.89 | 31.86 ± 2.47 |  |  |
| p | NS | NS | NS | NS | NS | 0.000 |  |

Note:
BMI, body mass index; NS, not significant.

and accurate results (*Seidell, Bakker & van der Kooy, 1990*; *Goran et al., 1996*; *Lifshitz et al., 2016*; *Andreoli et al., 2016*). However, it has also been reported that these methods have limited use, since they require special equipment, are expensive and time-consuming, and involve application-related difficulties in children (*Mei et al., 2002*). BMI is widely used to assess adiposity in the human body. Because weight gain is associated with growth of other body tissues rather than an increase in adipose tissue, BMI falls short of assessing patients in late childhood and adolescence. Therefore, the use of age and gender-specific BMI percentile is used as a rapid, non-invasive and easily accessible assessment method in this period (*Akridge et al., 2007*; *Haroun et al., 2005*; *Silveira et al., 2011*; *Soliman et al., 2011*).

Many studies have reported that obesity accelerates skeletal maturation and growth in childhood and early puberty (*Freedman et al., 2003*; *Slyper, 2006*; *Giuca et al., 2012*). The size of craniofacial structures in obese individuals also increases compared to those of normal weight (*Giuca et al., 2012*; *Ohrn et al., 2002*; *Ferrario et al., 2004*). With the increase

**Table 3 Soft tissue thickness measurements of different BMI percentile groups and inter-group comparisons.**

| | | | | | | Tukey's HSD | | |
|---|---|---|---|---|---|---|---|---|
| | Group | *n* | Mean | SD | SE | I–II | I–III | II–III |
| Glabella | Underweight | 24 | 5.34 | 0.62 | 0.12 | NS | 0.000 | 0.000 |
| | Normal weight | 24 | 5.49 | 0.74 | 0.15 | | | |
| | Overweight | 24 | 6.69 | 1.19 | 0.24 | | | |
| Nasion | Underweight | 24 | 5.23 | 1.05 | 0.21 | 0.021 | NS | NS |
| | Normal weight | 24 | 6.05 | 1.15 | 0.23 | | | |
| | Overweight | 24 | 5.69 | 0.84 | 0.17 | | | |
| A-Sn | Underweight | 24 | 15.61 | 2.11 | 0.43 | NS | NS | NS |
| | Normal weight | 24 | 14.59 | 1.47 | 0.30 | | | |
| | Overweight | 24 | 15.69 | 1.67 | 0.34 | | | |
| Labiale superius | Underweight | 24 | 11.93 | 1.70 | 0.34 | NS | 0.000 | 0.001 |
| | Normal weight | 24 | 12.55 | 2.03 | 0.41 | | | |
| | Overweight | 24 | 14.56 | 1.61 | 0.32 | | | |
| Stomion | Underweight | 24 | 5.27 | 1.64 | 0.33 | NS | 0.000 | 0.042 |
| | Normal weight | 24 | 6.14 | 1.56 | 0.31 | | | |
| | Overweight | 24 | 7.28 | 1.64 | 0.32 | | | |
| Labiale inferius | Underweight | 24 | 13.39 | 1.84 | 0.37 | NS | 0.003 | NS |
| | Normal weight | 24 | 14.11 | 1.38 | 0.28 | | | |
| | Overweight | 24 | 15.15 | 2.03 | 0.41 | | | |
| Labiomentale | Underweight | 24 | 10.33 | 0.92 | 0.18 | NS | NS | NS |
| | Normal weight | 24 | 11.08 | 1.39 | 0.28 | | | |
| | Overweight | 24 | 11.15 | 1.76 | 0.45 | | | |
| Pogonion | Underweight | 24 | 7.84 | 1.31 | 0.26 | NS | 0.000 | 0.017 |
| | Normal weight | 24 | 8.77 | 1.18 | 0.24 | | | |
| | Overweight | 24 | 9.91 | 1.66 | 0.33 | | | |
| Gnathion | Underweight | 24 | 6.77 | 1.15 | 0.23 | 0.045 | 0.000 | NS |
| | Normal weight | 24 | 7.72 | 1.13 | 0.23 | | | |
| | Overweight | 24 | 8.59 | 1.68 | 0.34 | | | |
| Menton | Underweight | 24 | 6.27 | 1.03 | 0.21 | NS | 0.000 | 0.013 |
| | Normal weight | 24 | 7.97 | 0.94 | 0.19 | | | |
| | Overweight | 24 | 7.86 | 1.17 | 0.23 | | | |

**Note:**
BMI, body mass index; HSD, honest significant difference; SD, standard deviation; SE, standard error; NS, not significant.

in obesity prevalence, epidemiological studies have been conducted to evaluate the possible effects of nutritional status on tooth development. They have shown that nutritional status may cause accelerations or delays in tooth development (*Mack et al., 2013*; *Kumar, Patil & Munoli, 2015*; *Bagherian & Sadeghi, 2011*).

As patients' demand for facial esthetic procedures increases, the orthodontic treatment paradigm shifts from the treatment of hard tissue to the soft tissue origin (*Cha, 2013*). Facial features cannot be predicted based on hard tissues alone; therefore, soft tissues

**Table 4 Soft tissue height measurements of different BMI percentile groups and inter-group comparisons.**

| | | | | | | Tukey's HSD | | |
|---|---|---|---|---|---|---|---|---|
| | Group | n | Mean | SD | SE | I–II | I–III | II–III |
| Sn-ULB | Underweight | 24 | 18.31 | 2.13 | 0.43 | NS | NS | NS |
| | Normal weight | 24 | 17.65 | 1.75 | 0.35 | | | |
| | Overweight | 24 | 17.48 | 3.12 | 0.63 | | | |
| LLB-LLV | Underweight | 24 | 9.04 | 1.35 | 0.35 | NS | NS | NS |
| | Normal weight | 24 | 9.77 | 1.15 | 0.29 | | | |
| | Overweight | 24 | 10.25 | 1.68 | 0.43 | | | |
| Sn-LLV | Underweight | 24 | 29.07 | 3.22 | 0.65 | NS | NS | NS |
| | Normal weight | 24 | 30 | 2.27 | 0.46 | | | |
| | Overweight | 24 | 30.36 | 4.58 | 0.93 | | | |

Note:
BMI, body mass index; Sn-ULB, subnasale-upper labial bottom; LLB-LLV, lower lip bottom-lower lip vermillion; Sn-LLV, subnasale-lower lip vermillion; HSD, honest significant difference; SD, standard deviation; SE, standard error; NS, not significant.

should also be considered during facial analysis (*Cha, 2013*). *Holdaway (1983)*, *Spradley, Jacobs & Crowe (1981)*, *Bell, Jacobs & Quefada (1986)*, *Owen (1984)*, and *Park & Burstone (1986)* are some of the many researchers emphasizing the importance of soft tissues in diagnosis.

The literature contains many studies evaluating soft tissue cephalometric norms for ethnic populations of different chronological ages (*Cha, 2013*; *Basciftci, Uysal & Buyukerkmen, 2003*; *Kalha, Latif & Govardhan, 2008*). In studies examining facial soft tissue thicknesses according to gender, it has been reported that these values are higher in men than in women (*Uysal et al., 2009*; *Hamdan, 2010*). Differences in facial soft tissues have also been noted in individuals with different skeletal malocclusions (*Kamak & Celikoglu, 2012*; *Utsuno et al., 2014*; *Pithon et al., 2014*). However, to the best of our knowledge, to date, no study has been conducted to investigate the effect of BMI on facial soft tissue thickness or height.

*Sahni et al. (2008)* suggested that wrinkles in the skin are associated with soft tissue thickness due to decreased skin elasticity and collagen thickness. Therefore, we eliminated age differences between the groups in our study.

*Arnett & Gunson (2004)* recommended that the patient be positioned in a relaxed lip position when assessing the soft tissue profile, since this position reveals the relationship of soft tissues with hard tissues without muscle compensation for dentoskeletal abnormalities. In previous studies, the relaxed lip position was used to standardize the method while taking cephalometric radiographs to accurately evaluate soft tissues. Consistent with these studies (*Uysal et al., 2009*; *Arnett & Gunson, 2004*), we used relaxed lip positions when taking cephalograms to assess soft tissue thicknesses accurately.

*Krebs et al. (2007)* reported that BMI and body fat ratio were proportional. Supporting this finding, we also determined that all soft tissue thickness measurements were higher in the overweight group, except for the nasion.

In our comparison of the underweight and overweight groups, we detected statistically significant differences in the thickness measurements at the glabella, labiale superius, stomion, labiale inferius, pogonion, gnathion, and menton. Differences in subcutaneous adipose tissue between individuals can affect facial soft tissue thickness. When examining the face, the cheek, cheek-eye, and chin-neck generally contain the most subcutaneous fat. These regions are also called the buccal fat pad, infraorbital region, and branches of the anterior neck, respectively (*Cha, 2013*). In our study, the soft tissue thicknesses at the gonion, pogonion, and menton corresponded to the main regions of subcutaneous fat on the face. The significance level of the difference between the underweight and overweight groups was more substantial compared to the remaining significant data ($p < 0.005$).

This study showed that patients with high BMI values had higher soft tissue thickness measurements. Although clinical changes are expected in the soft-tissue profile with bone and tooth movements, it seems reasonable to assume that the same movements in owerweight patients would produce less dramatic changes in the profile because of the increased thickness of the soft-tissue drape. Therefore, in high BMI patients that receive functional treatment, the effect of skeletal changes on the facial soft tissues may be mediated by a reduced soft tissue response, or if an extraction treatment protocol is applied, the effects on the profile and the lips may also be reduced (*Mankad et al., 1999*; *Attarzadeh & Adenwalla, 1990*).

We used lateral cephalometric radiographs to evaluate the soft tissue measurements. However, this method has been shown to have serious limitations compared to cone-beam computed tomography (CBCT), such as distortion, low reproducibility, and overlapping of bilateral craniofacial structures (*Lowe et al., 1995*). However, due to ethical concerns, we do not routinely record CBCT in patients receiving orthodontic treatment. There are studies in the literature that the position of the mandibular incisor will determine the position and shape of the lower lip (*Roos, 1977*). In our study, measurements were made without applying any orthodontic treatment to the patients, and lower incisor crowding or incisor angle measurements were not evaluated. Further studies may be planned the effect of incisor angle and incisor crowding on facial soft tissues.

## CONCLUSIONS

This study showed that adolescents with higher BMI had higher facial soft tissue thickness measurements. Therefore, the BMI can be used as a risk-free predictor of facial soft tissue thickness before orthodontic treatment, prior to or without any relevant radiographic documentation.

### Funding

The authors received no funding for this work.

### Competing Interests

The authors declare that they have no competing interests.

## Author Contributions

- Nurver Karsli conceived and designed the experiments, prepared figures and/or tables, authored or reviewed drafts of the article, and approved the final draft.
- Esra Tuhan Kutlu conceived and designed the experiments, performed the experiments, analyzed the data, prepared figures and/or tables, authored or reviewed drafts of the article, and approved the final draft.

## Clinical Trial Ethics

The following information was supplied relating to ethical approvals (*i.e.*, approving body and any reference numbers):

The ethics committee of the Faculty of Dentistry of Karadeniz Technical University approved this study (number: 2022/4, date: 07/09/2022).

## Data Availability

The raw data are available in the Supplemental Files.

## Supplemental Information

Supplemental information for this article can be found online at http://dx.doi.org/10.7717/peerj.16196#supplemental-information.

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
