# Peer review of "Effect of body mass index on soft tissues in adolescents with skeletal class I and normal facial height"

_PeerJ, doi:10.7717/peerj.16196_

## Round 0.1 · original submission · Major Revisions

The reviewers provided mixed reports for the study. At present no definitive decision can be made, since both reviewers expressed valid concerns regarding the rationale, as well as methodological limitations of the study. Please try to address all concerns adequately and resubmit the study.

Reviewer 1 ·

Basic reporting

no comment

Experimental design

This is a cross-sectional study on (probably) pretreatment lateral cephalograms. The main conclusion, being that a higher BMI is associated with thicker soft tissues, does not have any clinical significance or any diagnostic value, as it is a correlation that was expected to be statistically significant before study commencement. There are already similar studies in the literature that investigated this correlation (PMID: 29318946, 36758899). Therefore, I find the research question and the rationale for this study lacking. What would be a clinically relevant question, is the effect of BMI on the soft tissue changes following orthodontic treatment (for example after retraction of the upper teeth or 4 premolar extractions).

Validity of the findings

no comment

Reviewer 2 ·

Basic reporting

This paper presented about Facial soft tissue thick ness of Turkish population divided into Body Mass Index which measured the cephalogram. The author discussed about the differences of soft tissue thickness on mid sagittal plane in skeletal class I in each BMI. The Idea and Methodology is orthodox. However, lacking of explanations are found in this paper.

Experimental design

Materials and method
1. The author should add the detail of devices (model of X-ray system).
2. The author should describe the settings of cephalometric X-ray (include kv, sec, etc.). Film-tube distance is different by country.
3. This paper should add the definitions of measured landmarks. Because the reader of this journal is not orthodontists. There are differences of definition between anthropologists and orthodontists. For example, anthropologists recognized the gnathion as the menton.
4. The author should explain the magnify of cephalogram. Recalculation is essential to obtaining actual thickness. If the author did not recalculate, the author should perform.
5. In L.89, the author described “~ relaxed lip position with~”. Were the participants with disable to taking lip exclude? If so, the author should add this in excluding standards.
6. The formula of Intra and inter observer errors are essential. To show the reliability and repeatability of the author’s research. Please add the table of reliability and repeatability.

Validity of the findings

Discussion
1. The author discussed the differences of thickness and height in lower face region. To discuss this region, the author should discuss about the dentition of incisor region (incisor crowding, angle of incisor axis, etc.)

Annotated reviews are not available for download in order to protect the identity of reviewers who chose to remain anonymous.

---

## Round 0.2 · Major Revisions

Dear Authors,

1. The need of the study and 2. the interpretation/importance of the findings need to be clearly provided for the readers.

I understand that based on these findings, the clinicians have more prediction power regarding soft-tissue thickness, based on BMI. This is relevant only if a ceph is not available, e.g. prior to documentation or for radiation reduction purposes. This needs to be clearly reported in the Introduction. The following reference can be used: DOI: 10.1093/ejo/cjac002. Otherwise the treating doctor should perform individual patient measurements to make decisions and not use average measures of a sample, which this study provides.
Lines 64-66: "It will contribute more aesthetic results in treatment plans made by considering the relationship of BMI and soft tissues before orthodontic treatment, which will cause a change in facial appearance, like functional treatments."
This is a too strong argument. The response of soft-tissues in hard-tissue changes cannot be accurately predicted in an individual patient level.
Please revise the Introduction and the discussion parts based on the above comments and consider carefully all reviewer's comments to further improve the manuscript prior to resubmission.

Reviewer 1 ·

Basic reporting

1. Line 141: This argument is rather strong considering that the unreliability of the BMI is known issue among clinicians for both adults and children (example reference: PMID: 25503667). Reference No26 cannot support your argument. Please modify it or use a more accurate reference.

2. The intraclass correlation coefficients are still not properly reported. The provided supplementary file only provides paired statistic tests between the repeated measurements. No ICCs are reported and it is also not reported if the repeated measurements were performed by the same or another operator (or both). The repeated measurements are also missing from the file containing the raw data. Please provide proper supplementary files with ICC statistics. The ICCs would be better to be reported in a small table in the main manuscript.

Experimental design

1. Please define properly the distances used for the soft tissue thickness measurements. Apparently the distance between the soft tissue landmarks and their hard tissue counterparts was measured. This should be added in the materials and methods section. In the figure legend (and/or materials and methods) you should also report the landmark pairs for each measurement.

Validity of the findings

1. Lines 200-202: This study found a significant correlation between BMI and soft-tissue thickness. However, it was not discussed how BMI can actually be incorporated during diagnosis and treatment planning. If a lateral cephalogram is available before treatment, how would the BMI affect the clinicians' decisions? Please elaborate more in the discussion or retract this argument.

Reviewer 2 ·

Basic reporting

The comments reviewer pointed were almost corrected. However one point to revise is still remain.
The author add the name of anthropological landmarks in Figure 1. additionally, the author should add the table of definitions of these landmarks.

Experimental design

already commented previous submission.

Validity of the findings

already commented previous submission.

Additional comments

no comment.

---

## Round 0.3 · Minor Revisions

Please revise the title as: "Effect of body mass index on soft tissues in adolescents with skeletal Class I and normal facial height".

Line 67-69: "This study will contribute to the clinicians' ability to predict soft tissue thicknesses of patients based on body mass index before patient documentation or when it is not desired to take radiographs to reduce the radiation dose." Please revise as "This study will aid clinicians in predicting the soft-tissue thicknesses of patients based on body mass index, before patient documentation or when it is not desired to take radiographs for radiation reduction purposes."

Please report in the statistical analysis section the exact type of ICC used.

Lines 195-187: "Therefore, the patient who is planned for functional treatment, the skeletal effect may not be as expected due to soft tissue thickness, or if the clinician considering extraction-based treatment, there may be less effect on profile and lip support. [50-51]." Please revise as: "Therefore, in high BMI patients that receive functional treatment, the effect of skeletal changes on the facial soft tissues may be mediated by a reduced soft tissue response, or if an extraction treatment protocol is applied, the effects on the profile and the lips may also be reduced. [50-51]."

Conclusions: "This study showed that patients with higher BMI had higher soft tissue thickness measurements. Therefore, BMI can give the clinician an idea about soft tissues before orthodontic treatment." Please revise as "This study showed that adolescents with higher BMI had higher facial soft tissue thickness measurements. Therefore, the BMI can be used as a risk-free predictor of facial soft tissue thickness before orthodontic treatment, prior to or without any relevant radiographic documentation."

---

## Round 0.4 · Minor Revisions

Dear Authors,

There is only one issue that still remains unclear. The exact type of used ICC needs to be reported in the statistical analysis section. Please refer to the following article for an overview of the different ICC forms available: Koo TK, Li MY. A Guideline of Selecting and Reporting Intraclass Correlation Coefficients for Reliability Research. J Chiropr Med. 2016 Jun;15(2):155-63. doi: 10.1016/j.jcm.2016.02.012.

---

## Round 0.5 · accepted · Accept

All of the reviewers' comments have been addressed. The manuscript has been successfully revised and is ready for publication.